# Analyzing the Difference in the Length of Stay (LOS) in Moderate to Severe COVID-19 Patients Receiving Hydroxychloroquine or Favipiravir

**DOI:** 10.3390/ph15121456

**Published:** 2022-11-24

**Authors:** Bandar Alosaimi, Huda M. Alshanbari, Muath Alturaiqy, Halah Z. AlRawi, Saad Alamri, Asma Albujaidy, Aljawharah Bin Sabaan, Ahmed A. Alrashed, Ahmad Alamer, Fayez Alghofaili, Khaled Al-Duraymih, Abdulaziz J. Alshalani, Wael Alturaiki

**Affiliations:** 1Research Center, King Fahad Medical City, Riyadh Second Health Cluster, Riyadh 11525, Saudi Arabia; 2Department of Mathematical Sciences, College of Science, Princess Nourah bint Abdulrahman University, Riyadh 11671, Saudi Arabia; 3Department of Internal Medicine, Alzulfi General Hospital, Riyadh Second Health Cluster, Riyadh 11525, Saudi Arabia; 4Department of Clinical Pharmacy Service, Prince Mohammed bin Abdulaziz Hospital, Riyadh Second Health Cluster, Riyadh 11525, Saudi Arabia; 5Pharmacy College, Almaarifa University, Riyadh 11597, Saudi Arabia; 6Department of Pharmaceutical Services, Main Hospital, King Fahad Medical City, Riyadh Second Health Cluster, Riyadh 11525, Saudi Arabia; 7Department of Clinical Pharmacy, Prince Sattam Bin Abdulaziz University, Alkharj 11942, Saudi Arabia; 8Department of Pharmacy Practice and Science, College of Pharmacy, University of Arizona, Tucson, AZ 85721, USA; 9Department of Medical Laboratory Sciences, College of Applied Medical Sciences, Majmaah University, Majmaah 11952, Saudi Arabia; 10Main Laboratory and Blood Bank, Alzulfi General Hospital, Riyadh Second Health Cluster, Riyadh 11525, Saudi Arabia; 11Alzulfi General Hospital, Riyadh Second Health Cluster, Riyadh 11525, Saudi Arabia

**Keywords:** favipiravir, hydroxychloroquine, length of stay, COVID-19, SARS-CoV-2, effectiveness, statistical analysis

## Abstract

*Background*: The coronavirus 2019 (COVID-19) disease, caused by the severe acute respiratory syndrome coronavirus 2 (SARS-CoV-2) virus led to a global pandemic. HCQ and FPV were used early in the pandemic as a treatment modality for COVID-19. Various studies evaluated the HCQ and FPV effectiveness, based on the mortality endpoint and showed conflicting results. We hypothesize that analyzing the difference in the LOS as a significant endpoint would be of a major interest, especially for healthcare providers, to prevent a lengthy hospitalization and disease progression. *Methods*: This is a retrospective observational study, conducted via a medical chart review of COVD-19 patients who were admitted between April 2020 and March 2021 with a moderate to severe illness. The LOS endpoint was tested using the paired Wilcoxon signed-rank (WSR) model. Prior to using the WSR model, the balance between the HCQ and FPV groups, the propensity score matching, the LOS distribution, and the normality assumptions were tested. Two sensitivity statistical analyses were conducted to confirm the results (stratified log-rank test and U Welch test after transforming the LOS by the squared root values). *Results*: A total of 200 patients were included for the analysis: 83 patients in the HCQ group and 117 patients in the FPV group. Thirty-seven patients were matched in each group. The LOS data was positively skewed and violated the normality (Shapiro–Wilk *p* < 0.001) and had an unequal variance (Levene’s test, *p* = 0.019). The WSR test showed no statistical significance in the LOS endpoint, with a median of −0.75 days (95% confidence interval: −4.0 to 2.5, *p* = 0.629), in favor of the HCQ group (four days), in comparison to seven days of the FPV group. The WSR findings were further confirmed with the stratified log rank test (*p* = 740) and the U Welch test (*p* = 391). *Conclusions*: The study concluded that the HCQ and FPV treatments have a comparable effectiveness in terms of the LOS in the moderate to severe COVID-19 patients. This study highlights the importance of analyzing the LOS as a relevant endpoint, in order to prevent the costs of a lengthy hospitalization and disease progression. The current study also emphasizes the importance of applying the appropriate statistical testing when dealing with two-sample paired data and analyzing non-parametric data such as the LOS.

## 1. Introduction

The emergence of SARS-CoV-2 has caused a pandemic, with nearly half a billion confirmed cases of COVID-19, including over 6 million deaths, reported to the WHO [1]. In Saudi Arabia, as of April 2022, there have been more than 750 thousand confirmed cases with at least 9 thousand deaths. Although the majority of SARS-CoV-2 infections produce mild symptoms or are asymptomatic, some infected individuals require hospital care [2].

Although the safety and effectiveness of new COVID-19 investigational therapies continue to be evaluated, the U.S. FDA issued an Emergency Use Authorization (EUA) for the drugs paxlovid, molnupiravir, baricitinib and tocilizumab, which were shown to reduce to the recovery time [3]. To control an outbreak for a particular infectious disease, the discovery of new drugs appears to be the most effective way to treat patients. However, the time needed to develop and assess a new drug is lengthy.

There are many available antivirals that have proven to be successful with already established safety profiles, such as HCQ and FPV [4]. Furthermore, many agents have been recorded as having anti-SARS-CoV-2 effects, such as lopinavir and ritonavir, which were originally used to treat the HIV virus; remdesivir, an Ebola antiviral drug [5]; and dexamethasone. All have shown both improved outcomes and a reduced mortality [6]. HCQ has long been used for the treatment of malaria, and has become the standard treatment for COVID-19. HCQ can reduce the infection of SARS-CoV-2 by glicolizing the viral spike proteins and the human lung receptors, preventing the viral entry [7]. FPV has also been used in severe influenza cases. The drug can inhibit the RNA polymerase activity by both blocking the replication process and adversely acting on the genetic copying [8].

The Saudi protocol to treat patients with a confirmed COVID-19 has included HCQ and FPV, since the beginning of the pandemic [9]. However, few studies have been reported, comparing the treatment modality, based on multiple endpoints, to evaluate the effectiveness of HCQ and FPV. In addition, there is insufficient comparative data on the effectiveness of treatment analyzing the difference in the LOS in hospitals as a significant endpoint. For these reasons, this study, aimed to evaluate the effectiveness of HCQ and FPV, among moderate to severe COVID-19 Saudi patients, by analyzing the difference in the LOS, using the paired Wilcoxon signed-rank (WSR) model, in order to prevent the costs of a lengthy hospitalization and disease progression.

## 2. Results

We identified 278 patients admitted to the hospitals, of which 127 patients received HCQ, and 130 patients received FPV. All medical records were screened for eligibility to include only adults (≥18 years old) with a confirmed diagnosis of moderate to severe COVID-19 and who received treatment of either HCQ or FPV. A total of 200 patients were included for the analysis (83 patients in the HCQ and 117 patients in the FPV groups). Prior to matching, the mean (SD) age for the HCQ group was 49.48 (15.46), compared to 62.25 (15.9) in the FPV group (*p* < 0.001). Male patients were in the majority in the HCQ (71.1%) and FPV groups (60.7%) (*p* = 0.171). More Saudi nationals were in the FPV group (81.2%), compared to 36.1% who received the HCQ treatment. There was no difference in the body mass index (kg/m^2^) between the two groups, with a mean (SD) of 30.2 (7.1) for the HCQ group and 28.8 (3.1) for the FPV group (*p* = 0.065). Patients who received HCQ had a more severe illness (16.1%) and received fewer steroids (7.2%), compared to the FPV group with 11.1% and 27.4% of patients, respectively. Diabetes and hypertension were common in the two groups but not statistically significant. Cardiovascular disease was more frequent in the HCQ group (9.6%) versus 1.7% in the FPV group. There were more patients with asthma in the HCQ group: 19.3% versus 4.3% in the FPV group. Chronic kidney disease was more prevalent in the HCQ group (6.0%) than in the FPV group (0.0%), as shown in Table 1.

Once matched, the baseline characteristics were adequately balanced, with the SMD values < 0.2. The balance cohort is shown in Table 1. A total of 37 patients were matched in each group. The mirror plots for the propensity score distributions between the treated (FPV = 1) and control (HCQ = 0), indicated an adequate balance (Appendix A). The Love’s plot provided a visual representation of the SMD values for the covariates of interest across the multiple inputted datasets (Appendix A).

The LOS distribution was positively skewed with a tail to the right (skewness of 1.7 in the HCQ group and 1.1 for the FPV group). The result of the Shapiro–Wilk test was a significant *p*-value < 0.001, as was the Levene’s test, (*p*-value = 0.019), indicating an unequal variance. In the matched sample, the LOS median (IQR) for the HCQ was four days (0 to 10), and seven days (4 to 9) for the FPV, with no statistical difference, based on the Wilcoxon signed-rank test between the two groups (−0.75; 95% confidence interval (CI), −4.0 to 2.5, *p*-value = 0.629). A visual representation of these findings is shown in Figure 1A.

The Welch U test was also not significant, with a *p*-value *=* 0.391. The square root transformation made the LOS outcome close to the normal distribution; however, it did not improve the variance of the variable (Figure 1B).

The Kaplan–Meier model showed that the median time to discharge was five days (95% CI, 2 to 10) for the HCQ group and seven days for the FPV group (95% CI, 5 to 10), with no statistical significance between the two groups (*p*-value = 0.740), as shown in Figure 2. Two deaths occurred in the HCQ group and no deaths occurred in the FPV group, *p*-value = 0.493 (Table 2). None of the patients included in this study required mechanical ventilation during their hospital stay.

## 3. Discussion

The study provided an opportunity to advance the knowledge of analyzing the effect of HCQ versus FPV on the LOS, as a surrogate for the marker effectiveness. We observed that both drugs showed a comparable median LOS in moderate to severe COVID-19 patients. There have been few head-to-head comparative quantitative studies that investigated the clinical outcomes of HCQ and FPV [4,10]. Previous studies have only focused on laboratory tests, such as the clearance of SARS-CoV-2 by the PCR, and radiological improvements which may not reflect the clinical recovery [4,10]. The little data in the literature are controversial, and there is no general agreement about the difference in the LOS and mortality outcomes between the HCQ and FPV groups.

The most likely explanations of the contradictory findings between both groups are poor study designs, heterogeneity of the study populations, and power of the study, small sample sizes, early vs. late initiation of the antiviral treatment, diversity of dosing, therapy regimen and duration of the treatment plan [10,11]. There are several studies comparing the LOS of an HCQ-based regimen to a control arm. In a randomized control trial, Abd-Elsalam et al., reported that the mean LOS did not differ between the HCQ and the supportive groups, with 11.04 days and 11.27 days, respectively (*p*-value = 0.52) [12]. Likewise, an open-label RCT study, compared the LOS of moderate COVID-19 cases in patients receiving HCQ, lopinavir/ritonavir, and lopinavir/ritonavir plus interferon, and found that they were identical, with a mean LOS of 11, 12, 11 days, respectively (*p*-value = 0.20) [13]. In contrast, a meta-analysis demonstrated that antiparasitic drugs, including HCQ, were associated with a shorter length of hospital stay, in comparison with the standard of care in a control group who were taking a placebo (*p*-value < 0.05) [14]. Conversely, Alamer et al., found that the FPV group had a shorter median time to discharge among the moderate COVID-19 cases, than the supportive-care group of nine days (95% CI, 9 to 10) versus 11 days (95% CI, 10 to 12) [15].

As many studies showed inconsistencies in the LOS outcomes, a debate about mortality with HCQ and FPV has also created interest within the scientific community. Studies reported varying findings, ranging from large decreases in mortality [16], no changes in mortality [17] and moderate to large increases in mortality [18]. One study raised enough red flags to stop the use of HCQ, with the preliminary analysis revealing that the risk unexpectedly outweighed the benefits [19]. The reported adverse events are an increased all-cause mortality, ICU admission, mechanical ventilation, QTc interval prolongation, and ventricular arrhythmias [19]. The UK Medicines and Healthcare Products Regulatory Agency requested that the data be scrutinized and further reviewed. Unfortunately, their analysis was prompted by the retractions of high-profile papers in journals, such as *The Lancet* [20].

Based on the current data available, the use of HCQ and FPV was limited because numerous anti-COVID-19 drugs were found to be potentially useful, with various levels of supporting evidence [5,6,21]. We noted in our study that the mean age in the unmatched sample was 49 years in the HCQ group and 62 in the FPV group. This finding may be related to the prescribers’ preferences or to institutional policies that considered HCQ to be unsafe for the elderly, given the multiple reports of safety-related concerns [20,22,23]. The age discrepancy favoring the young for HCQ is not unique to this study. The study by Alotaibi et al., conducted in Saudi Arabia, reported a similar mean age (mean = 49.6 with SD = 12.6), which was statistically significant (*p*-value < 0.001), compared with the FPV group (mean = 54.8 with SD = 14.9) [4]. The higher age range in the FPV may also be related to the relative safety of the medication, as more information has emerged since its approval [24].

As many COVID-19 treatments failed to show mortality benefits [25], it is critical to evaluate the LOS endpoint, as COVID-19 patients can put a strain on the healthcare system [26]. A caveat is that much of the published literature failed to report meeting the statistical test assumptions, giving rise to the question of the validity of the statistical tests conducted [27,28]. In the case of COVID-19, this may be related to rapid research reporting and dissemination, which can lead to the misinterpretation of the data [29].

In order to analyze our primary endpoint accurately, our study addressed the significant baseline characteristics imbalance we encountered, using propensity score matching. In our propensity score model, important variables of interest were included. The procedure reduced our sample size (*n* = 37 in each group) but ensured the comparability between the two groups. Figure 1A shows the probability density function for the outcome of the LOS which was positively skewed with a tail to the right, with few outliers for the HCQ group.

Statistical test assumptions are often violated when analyzing the difference in the LOS between two samples. Failure to test for the LOS distribution can lead to selecting an inappropriate statistical test. For instance, a literature review from five major emergency medicine journals (data from 2004 to 2007) with a focus in the emergency department (ED) LOS endpoint, revealed that 80% of the included studies did not perform a test of normality on the LOS data and 43% of the studies failed to perform the appropriate non-parametric tests [30]. Applying a parametric test to the non-normally distributed data may result in a type II error-failure to detect the difference. Chazard et al., a total of 12 statistical tests commonly used to analyze the LOS data were evaluated. In the case of a balanced sample (equal sizes of *n*), it is advisable to use either the Wilcoxon test, or the Kruskal–Wallis test (known as pure shift models) or Student’s t-test with a log or rank transformation, when comparing the LOS non-parametric data. These tests give high power with relative efficiency, compared with Student’s t-test (Student’s t-test is very conservative, with a type I error < 0.5%) [31]. The log-rank test was not appropriate in the simulated study as a statistical test for the LOS, as it gave unacceptable high α errors. In our study, the Wilcoxon test agreed with the stratified log-rank test that there was no difference between the two groups in the LOS endpoint. However, the simulated study did not address the problem of the unequal variance as a test assumption for the Wilcoxon test. The unequal variance could have affected the accuracy of the Wilcoxon test. Therefore, we decided to modify the LOS by the square root transformation. The square root transformation made the LOS outcome close to a normal distribution but did not improve the variance of the variable. A re-test for our hypothesis, using the Welch U test with the log LOS also produced non-significant results (*p*-value = 0.391).

Finally, several important limitations need to be considered. The study was retrospective in nature. We did not evaluate the safety profile of HCQ, despite the existing an international debate about the considerable cardiovascular adverse events reported. As the study data was collected early in the pandemic, it did not capture the effect of the vaccine status, improvements in the treatment protocols, nor emergence of COVID-19 variants. Moreover, we did not examine the effect of removing the outliers from the LOS outcomes, since this would have reduced the sample size and created unequal sizes, calling into question the validity of our statistical tests. The numbers of patients and controls were also relatively small, which may have affected the power of the study and the generalizability of the results. The distribution of the LOS data could have been improved by increasing the sample size or selecting a statistical test that can achieve a greater power while controlling the type I errors. Finally, despite the use of the propensity score matching to account for the confounders, the residual confounding cannot be entirely ruled out.

## 4. Methodology

### 4.1. Study Design and Setting

The research was a retrospective observational study conducted via a medical chart review of two Riyadh Second Health Cluster hospitals: Prince Mohammed Bin Abdulaziz Hospital (500 beds) and Al-Zulfi General Hospital (150 beds). The Institutional Review Board at the tertiary specialty referral hospital in the cluster, known as King Fahad Medical City in Riyadh, approved the study (IRB 22-175).

### 4.2. Selection Criteria

A list of COVID-19 patients admitted to the two hospitals between April 2020 and March 2021 and who had received either FPV or HCQ treatments, was obtained. To minimize bias, a random-selection process was implemented while screening for eligibility. Patients with a confirmed diagnosis of COVID-19 by a PCR had an equal opportunity to be selected if they were at least 18 years old, received either FPV or HCQ treatment and had been admitted with a moderate to severe infection, as classified by the World Health Organization (WHO). Patients with mild or critical infections, classified by the WHO, who had an incomplete medical record, or were admitted outside of the study duration, were excluded from the study.

### 4.3. Definitions

We classified patients according to the WHO definition into moderate or severe [32]. Patients with non-severe pneumonia, who were breathing normally without supplemental oxygen, were classified as moderate patients. A severe COVID-19 disease was defined as the development of fever, in addition to one or more of the following symptoms: respiratory rate ≥ 30/min, dyspnea, respiratory distress, SpO_2_ ≤ 93% on room air, PaO2/FiO2 ratio < 300 or lung infiltrate > 50% of the lung field within one to two days. The length of stay (LOS) was defined as the date of admission minus the date of discharge and was reported in days.

### 4.4. Study Outcomes

The primary endpoint was the difference in the LOS between the two treatments. The second endpoint was the mortality difference between the two treatments.

### 4.5. Baseline Information Collection

The demographic data that included age, sex, body mass index (kg/m^2^), ethnicity, comorbidities, steroid use, length of stay, respiratory assistance and patients’ outcomes, were extracted. Data quality during the data collection was periodically checked to assess the consistency and to resolve any discrepancies.

### 4.6. Sample Size Calculation and Statistical Analysis

We calculated the sample size, based on the assumption that the LOS distribution would violate the normality distribution. The sample size was calculated using the MKpower package [33]. The package is written in R language with various functions to calculate the sample size for the Wilcoxon rank sum and signed rank tests. In our parameter inputs, we assumed a mean difference of −2 and a standard deviation (SD) of 2 between the two treatment groups for a 0.05 α significance level (type 1 error rate). Based on our calculations, a sample size of 20 patients in each group was needed to achieve a power (beta) > 80%.

Continuous variables with normal distributions were presented as mean ± (standard deviation [SD]) and the statistical significance between the two groups was tested using Student’s t-test. The Chi-squared test and the Fisher exact test were used to compare the categorical data. There was a few missing data points in the dataset (such as the body mass index), due to the nature of the study (observational and dependent on a chart review). In order to perform an adequate analysis for the propensity score matching, a complete dataset is required to calculate the propensity scores for each case. For the main analysis, the missing data were first inputted using the multivariate imputation by chained equations (MICE), based on the assumption that the data were missing at random (five imputations with 10 iterations). To induce a balance between the two treatment groups, the treatment assignment was modelled, using a logistic regression model that included the measured pre-treatment covariates of interest. The log transformations were applied to the continuous variables for modeling. Within the datasets, the nearest neighbor method with a caliper of 0.2, were the specifications for the propensity scores matching procedure. Less than a 0.2 absolute standardized mean difference (SMD) indicated an adequate balance on the measured covariate. Love’s plot for the covariate balance and mirror plots for the propensity score distributions were generated (Appendix A).

The null hypothesis (H0) for the LOS outcome stated that there was no difference between the medians or means of the LOS of the two treatment groups. The analysis was carried out using only matched data. The LOS was presented as the median with interquartile ranges (IQRs). The normality was checked for the LOS outcome using the Shapiro–Wilk test and the Q-Q plots. An equal variance assumption was tested using Levene’s test. The paired Wilcoxon signed-rank test [34] was chosen to test the median differences in the LOS between the two treatment groups to account for the matched data [13]. Two sensitivity statistical analyses were conducted to confirm the results. First, the Welch U test was used, assuming an unequal variance with the square root transformation of the LOS, to test the difference between the two means. All days were shifted by one day for this test. The second test was using the survival analysis. For the second method, a Kaplan–Meier model was used to estimate the median time to discharge. Patients who were discharged were considered as events. To test for the significance in the survival curves, a stratified log-rank test was used for the matched pairs [35].

### 4.7. Software Used

R Core Team (2021) software (R Foundation for Statistical Computing, Version 4.1.2, Vienna, Austria) was used. The following R packages were used to generate results: ggplot2 [36], matchthem [37], cobalt [38], MKpower [33], mice [39] and survival [40].

## 5. Conclusions

Our study concluded that HCQ and FPV have a comparable effectiveness, in terms of the LOS in moderate to severe COVID-19 patients. Although not statistically significant, the median interquartile range of the LOS for patients treated with HCQ was nearly half of the LOS time for patients treated with FPV. The study highlights the importance of healthcare providers analyzing the LOS as a relevant endpoint to reduce the costs of a lengthy hospitalization and disease progression. The current study also emphasizes the importance of applying appropriate statistical testing when dealing with non-parametric data, such as the LOS.

## Figures and Tables

**Figure 1 pharmaceuticals-15-01456-f001:**
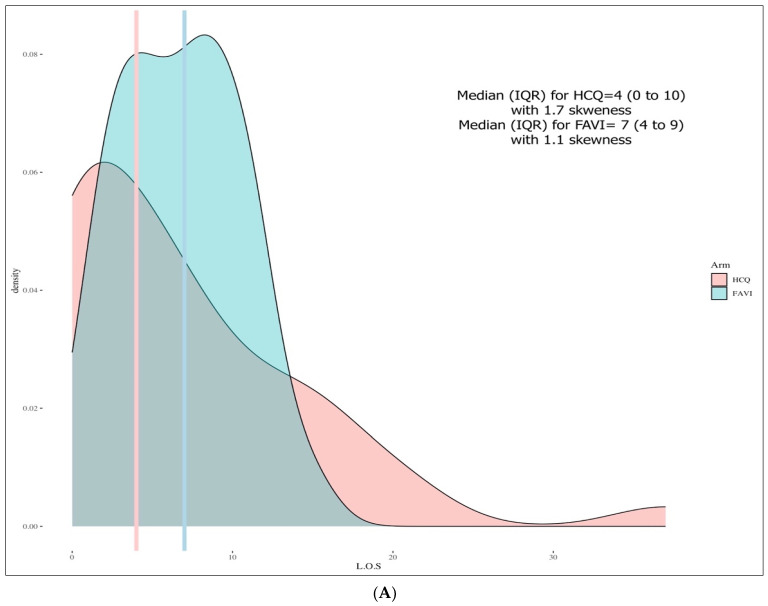
(**A**). Length of stay (LOS) density plot for the matched sample. Median presented for each group with the interquartile range (IQR). (**B**). square root length of stay (LOS) + 1-day density plot for the matched sample. Mean with the standard deviation (SD) presented for each group. HCQ = hydroxychloroquine. FAVI = Favipiravir. Vertical line for Figure 1A indicates median and Figure 1B are mean.

**Figure 2 pharmaceuticals-15-01456-f002:**
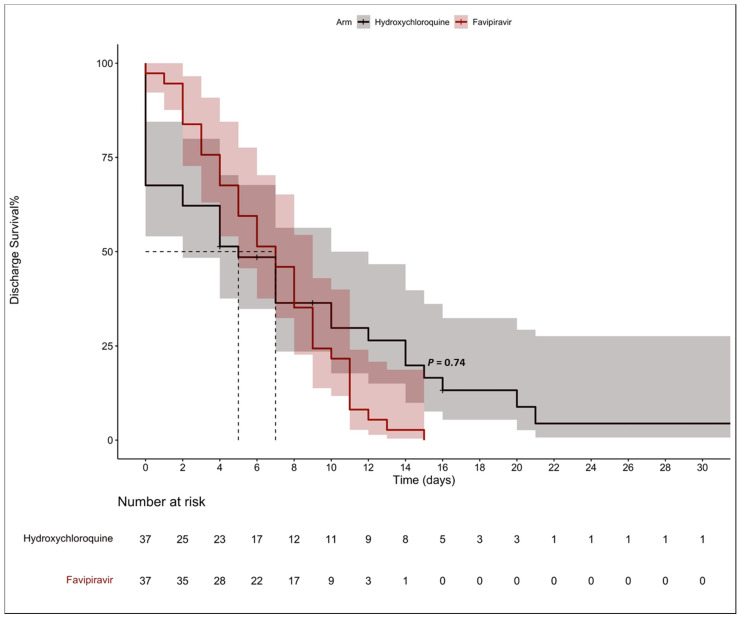
Kaplan–Meier for the median time to discharge in the matched sample. Dashed lines for the median. Decrease in survival indicates the discharge events.

**Table 1 pharmaceuticals-15-01456-t001:** Baseline characteristics (before and after the propensity score matching with the absolute standardized mean difference).

Variable	Total (*n* = 200)	HCQ (*n* = 83)	FPV (*n* = 117)	*p*-Value	Unadjusted Absolute SMD	HCQ (*n* = 37)	FPV (*n* = 37)	*p*-Value	Adjusted SMD
Age (years), mean (SD)	56.95 (16.9)	49.48 (15.46)	62.25 (15.9)	<0.001	0.8872	54.08 (17.49)	57.38 (16.22)	0.403	0.188
Male, *n* (%)	130 (65.0)	59 (71.1)	71 (60.7)	0.171		26 (70.3)	27 (73.0)	1.000	0.0417
Saudi nationality, *n* (%)	125 (62.5)	30 (36.1)	95 (81.2)	<0.001	0.4505	21 (56.8)	22 (59.5)	1.000	0.0487
Ethnicity, *n*(%) a				<0.001	NA			NA	NA
Middle Eastern	136 (68.0)	37 (44.6)	99 (84.6)			25 (67.6)	25 (67.6)		
African	19 (9.5)	15 (18.1)	4 (3.4)			5 (13.5)	3 (8.1)		
South/Southeast Asia	43 (21.5)	29 (34.9)	14 (12.0)			7 (18.9)	9 (24.3)		
Central Asia	1 (0.5)	1 (1.2)	0 (0.0)			0 (0.0)	0 (0.0)		
Unknown	1 (0.5)	1 (1.2)	0 (0.0)			0 (0.0)	0 (0.0)		
BMI (kg/m^2^)	29.40 (5.2)	30.2 (7.1)	28.83 (3.1)	0.065	0.3039	28.89 (4.20)	28.58 (2.91)	0.717	0.0650
Severe WHO classification, *n* (%)	27 (16.1)	14 (27.5)	13 (11.1)	0.015	0.1009	5 (13.5)	4 (10.8)	1.000	0.0208
Steroid, *n* (%)	38 (19.0)	6 (7.2)	32 (27.4)	0.001	0.2012	4 (10.8)	7 (18.9)	0.513	0.0587
Diabetes, *n* (%)	69 (34.5)	34 (41.0)	35 (29.9)	0.142	0.1105	8 (21.6)	9 (24.3)	1.000	0.0374
Hypertension, *n* (%)	82 (41.0)	33 (39.8)	49 (41.9)	0.877	0.0212	14 (37.8)	12 (32.4)	0.808	0.0529
Cardiovascular disease, *n* (%)	10 (5.0)	8 (9.6)	2 (1.7)	0.027	0.0793	1 (2.7)	2 (5.4)	1.000	0.0105
Asthma *n* (%)	21 (10.5)	16 (19.3)	5 (4.3)	<0.001	0.1500	5 (13.5)	4 (10.8)	1.000	0.0365
Chronic kidney disease, *n* (%)	5 (2.5)	5 (6.0)	0 (0.0)	0.026	0.0602	0 (0.0)	0 (0.0)	1.000	0.0000

SD: Standard Deviation. SMD: standardized mean difference. BMI: body mass index. WHO: World Health Organization. Cardiovascular disease included the following: myocardial infarction, ischemic heart disease or stroke. (a): was not included in the propensity score model. Note: Multiple imputation using the multivariate imputation by the chained equations for the missing variables (only 0.5% in the BMI and severity (16%) were missing). SMD values ere the preferred method to evaluate the covariate balance. Absolute SMD values < 0.2 indicated an adequate balance. Log transformation used age and the BMI for the propensity score modelling.

**Table 2 pharmaceuticals-15-01456-t002:** Clinical outcomes in the matched sample.

Matched Sample	HCQ (*n* = 37)	FPV (*n* = 37)	*p*-Value
Outcome			
LOS (days), median (IQR) ^a^	4 (0 to 10)	7 (4 to 9)	*p* = 0.629
Transformed LOS (days), mean (SD) ^b^	2.4(1.4)	2.7 (0.72)	*p* = 0.391
Median time to discharge (95% CI) ^c^	5 (2 to 10)	7 (5 to 10)	*p* = 0.740
Death, *n*(%) ^d^	2 (5.4)	0 (0.0)	*p* = 0.493

^a^ Paired Wilcoxon signed rank test. Pseudo (median) difference was −0.75 (95% confidence interval: −4 to 2.5). ^b^ Transformation by the square root. Welch U test was conducted. Mean difference was −0.21 (95% confidence interval: −0.70 to 0.28). ^c^ Kaplan–Meier estimates. *p* value was calculated using the stratified log rank test. ^d^ Fisher exact test was performed.

## Data Availability

Data is contained within the article and Appendix A.

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
