# Peer review of "Analyzing the Difference in the Length of Stay (LOS) in Moderate to Severe COVID-19 Patients Receiving Hydroxychloroquine or Favipiravir"

_pharmaceuticals, 2022, doi:10.3390/ph15121456_

Round 1

Reviewer 1 Report

This study is well designed and proper;y carried out. The statistical analyses are also appropriate and the results properly discussed. However, the relevance of the study is not obvious apart from the fact that it will add to the number (for statistics purposes) of studies which show that LOS could be used as a relevant endpoint in order to reduce the costs of lengthy hospitalization and disease progression. One major strength of the paper is demonstrating the appropriate statistical testing that is needed when dealing with non-parametric data such as LOS.

Major comment: I will advice that you re-analyse your data to bring out something additional than looking at the effect of HCQ versus FPV on the LOS as a surrogate for marker effectiveness.

Minor comment: Grammar should be checked. e.g. first sentence of Section 2.2. Consider including statistical analysis as a keyword.

Author Response

We would like to thank the reviewer for the positive feedback and comments. Please find the attached file .

Reviewer 2 Report

My comments and suggestions for authors as attached. 

Author Response

We would like to thank the reviewer for his positive feedback and comments. Please find the attached file 

Round 2

Reviewer 1 Report

My concerns for content have been properly addressed.

Author Response

We have addressed all the comments and suggestions raised by ref.1. Many thanks for the reviewer for this positive response.

Reviewer 2 Report

Well done ..good

Author Response

We have addressed all the comments and suggestions raised by ref.2. Many thanks for the reviewer for this positive response.